# Function in Cancer Patients: Disease and Clinical Determinants

**DOI:** 10.3390/cancers15133515

**Published:** 2023-07-06

**Authors:** Evelyn S. Qin, Blair Richards, Sean R. Smith

**Affiliations:** 1Department of Rehabilitation Medicine, University of Washington, Seattle, WA 98195, USA; eqin@uw.edu; 2Michigan Institute for Clinical Health Research, University of Michigan, Ann Arbor, MI 48109, USA; blairr@med.umich.edu; 3Department of Physical Medicine and Rehabilitation, University of Michigan, Ann Arbor, MI 48109, USA

**Keywords:** cancer, cancer type, functional outcomes, PROMIS, impairments, cancer function, physical function, PROMIS Cancer Function Brief 3D Profile, cancer rehabilitation

## Abstract

**Simple Summary:**

Day-to-day function in people with a history of cancer is incredibly important. It tells us what symptoms need to be treated, helps providers refer patients for rehabilitation, and helps us understand who is at risk of losing function. Our study looked at over 300 people with many different cancers, from five cancer centers in the United States. We found that people with active brain, sarcoma, prostate, and lymphoma cancers had the lowest function among people who were receiving rehabilitation. When cancer was cured, function was not related to the type of cancer someone had. Also, older people, overweight people, and people with non-cancer issues (like arthritis) had lower function. The results are the first of its kind to be reported and can lead to better decision-making for oncologists referring patients to rehabilitation care.

**Abstract:**

Patients with cancer often experience changes in function during and after treatment but it is not clear what cancer types, and associated clinical factors, affect function. This study evaluated patient-reported functional impairments between specific cancer types and risk factors related to disease status and non-cancer factors. A cross-sectional study evaluating 332 individuals referred to cancer rehabilitation clinics was performed at six U.S. hospitals. The PROMIS Cancer Function Brief 3D Profile was used to assess functional outcomes across the domains of physical function, fatigue, and social participation. Multivariable modeling showed an interaction between cancer type and cancer status on the physical function and social participation scales. Subset analyses in the active cancer group showed an effect by cancer type for physical function (*p* < 0.001) and social participation (*p* = 0.008), but no effect was found within the non-active cancer subset analyses. Brain, sarcoma, prostate, and lymphoma were the cancers associated with lower function when disease was active. Premorbid neurologic or musculoskeletal impairments were found to be predictors of lower physical function and social participation in those with non-active cancer; cancer type did not predict low function in patients with no evidence of disease. There was no differential effect of cancer type on fatigue, but increased fatigue was significantly associated with lower age (0.027), increased body mass index (*p* < 0.001), premorbid musculoskeletal impairment (*p* < 0.015), and active cancer status (*p* < 0.001). Anticipatory guidance and education on the common impairments observed with specific cancer types and during specific stages of cancer care may help improve/support patients and their caregivers as they receive impairment-driven cancer rehabilitation care.

## 1. Introduction

Over 16 million individuals are affected by cancer and its treatment in the United States [1]. As treatment and survival for cancer improves, a greater proportion of these individuals will develop early and late effects of their cancers and treatment that reduce function, including musculoskeletal pain, neuropathies, pain, and fatigue that may result in substantial motor and cognitive impairments and affect social participation [2,3,4,5]. As a result, individuals with cancer often experience reduced function in the home and community, and worsening psychosocial well-being [6,7]. What remains unknown, however, is the effect of different cancer types and non-cancer factors on a person’s function and how this may necessitate rehabilitation interventions to improve function and quality of life.

In 2019, the National Cancer Institute (NCI) identified evidence gaps in cancer survivorship care, which included a need to evaluate symptoms, functional status, and comorbid conditions, as well as the lack of routine capture of common data elements using validated core outcome instruments, including patient-reported outcomes (PROs) [8]. Function is incredibly important to people with a history of cancer—a person’s functional status can inform treatment decision making, and help providers identify patients needing rehabilitation to improve independence and ameliorate symptoms. Additionally, there is a large evidence gap as far as what cancers are associated with lower function, as well as non-cancer clinical factors that lower function. This stymies patient care, and understanding how function relates to cancer is essential to inform research.

There is, unfortunately, no agreed-upon standard for measuring function in cancer patients. PROs have been shown to reduce morbidity, mortality, and hospital utilization in cancer patients when applied electronically; early capture of symptoms may lead to earlier intervention [9,10]. The Patient Reported Outcome Measure Information System (PROMIS) Cancer Function Brief 3D Profile is a psychometrically tested patient-reported outcome measure tool developed for evaluating function in outpatient cancer rehabilitation patients [11,12].

Because the pathology and management of cancers vary widely based on the specific diagnosis and treatment options, there is a critical need to understand the extent of functional impairments of patients with various cancers, and the independent and additive effects of cancer type on function. In this study, we sought to evaluate differences in physical function, fatigue, and social participation between cancer types. Patient factors contributing to any differences in these functional outcomes, including age, body mass index (BMI), and premorbid non-cancer impairments, were also assessed. These findings may be important in facilitating impairment-driven cancer rehabilitation and care.

## 2. Methods

This cross-sectional study evaluated individuals with cancer referred to physical medicine and rehabilitation (PM&R) cancer rehabilitation clinics at six NCI and/or American College of Surgeons Commission on Cancer-affiliated hospitals between March 2018–March 2019. The primary clinical outcome examined was how cancer type impacts physical function, fatigue, and social participation domains of function. Further evaluation examined any interaction between cancer type and variables, including active disease status (no evidence of disease or not), BMI, pre-morbid history of neurologic or musculoskeletal impairments (e.g., history of a stroke with residual neurologic deficits, history of non-cancer amputation, or severe arthritis), and age.

Institutional Review Board (IRB) approval was obtained at each site, and written consent was acquired if required by the site’s IRB. Data quality, storage, and monitoring were performed through the coordinating site.

### 2.1. PROMIS Cancer Function Brief 3D Profile

The PROMIS Cancer Function Brief 3D Profile was administered to patients at all the study sites. The 12-item, psychometrically validated measure assesses physical function, fatigue, and social participation. It was developed as an easy-to-administer outcome measures instrument to assist with the delivery of cancer rehabilitation services [12]. Detailed methodology and validation of the measure have been described previously [11,13].

### 2.2. Participants

Participants were from a convenience sample of patients who presented for initial or follow-up evaluation in the cancer rehabilitation clinics with PM&R physicians. Inclusion criteria involved having a history of cancer, only one type of cancer, and being greater than 18 years of age. Patients limited by cognition, communication, and lack of English proficiency were excluded from the sample. Participants were further divided into groups based on cancer types (breast, colorectal, gynecologic, history of allogeneic bone marrow transplant (BMT), leukemia without allogeneic BMT, lung cancer, lymphoma without allogeneic BMT, melanoma, multiple myeloma, prostate, renal, sarcoma, and thyroid). Patients with multiple cancer types were excluded since the focus of this study was to measure the independent effects of cancer type on function.

### 2.3. Statistical Analysis

Univariable comparisons between cancer types were performed using the Fisher exact test for categorical variables and analysis of variance (ANOVA) for continuous variables. Only cancer types with greater than 10 cases and at least 5 cases within each cancer status group (active and non-active) were included in analyses. Multivariable analysis was conducted and included potential confounders identified a priori with examination for interaction between cancer type and active status. If interaction was indicated between cancer type and active status, subset analysis was conducted within the active and non-active cancer status groups. If the overall effect for cancer status was statistically significant, post-hoc analysis of least squares (LS) means (with 95% confidence intervals [CI]) was conducted to test between group differences. The potential confounders identified included: (1) cancer status (active versus non-active), (2) age, (3) body mass index (BMI), (4) premorbid neurological impairment, and (5) premorbid neurologic conditions. A 2-tailed *p* value of <0.05 was considered statistically significant without adjustment for multiple hypotheses testing. Statistical Analysis Software (SAS) 9.4 (Cary, NC, USA) was used for all analyses.

## 3. Results

A total of 332 patients were included in the analysis after removing cancer types with less than 10 cases (or less than 5 cases in the active or non-active status groups). Within this sample, 319 physical function scales, 325 social participation scales, and 327 fatigue scales were completed. The number discrepancy is because some patients completed only one or two of the three domains; incomplete questionnaires were not included in the results. Cancer diagnoses meeting the inclusion criteria included breast cancer (209), sarcoma (28), primary brain (26), gynecologic (23), prostate (19), colorectal (15), and lymphoma without allogeneic BMT (12).

The results showed that cancer had an independent effect on physical function amongst certain tumor types (brain, sarcoma, prostate, lymphoma), and that age, BMI, and non-cancer musculoskeletal factors also were associated with lower function. The details of our results across three functional domains, divided between patients with active and without active disease, are described henceforth.

### 3.1. Patient Demographics and Cancer Type

Baseline characteristics of age, BMI, gender, active disease status, premorbid neurological impairment, and premorbid musculoskeletal impairment were evaluated among the different cancer types (Table 1). There was a significant association between cancer type and age (*p* < 0.001), and between cancer type and active disease status (*p* = 0.004). Prostate cancer had the highest mean age (73.8 ± 8.6 years), and sarcoma had the lowest mean age (47.9 ± 15.3 years). There was no significant association between cancer type and BMI, gender, premorbid neurologic impairment, or premorbid musculoskeletal impairments.

### 3.2. Physical Function

Table 2 displays findings of the multivariable models that tested for interactions between type of cancer and cancer activity status in all the functional domains. A differential effect was found between cancer type and cancer status for physical function (*p* = 0.007). The differential effect remained significant when covariates were included as well (*p* = 0.03). Subset models within the active and non-active cancer groups were run to explore the differential effect. Physical function was associated with overall cancer type within the active disease subgroup (*p* < 0.001) but was not associated with cancer type within the non-active disease subgroup analyses (*p* = 0.340). Compared to breast cancer, all active cancer types except for colorectal and gynecologic cancer had a difference detected at the *p* < 0.05 level, with brain tumor having the largest difference (*p* < 0.001). For non-active disease, higher BMI (*p* < 0.001), and increased premorbid neurological (*p* = 0.002) and musculoskeletal impairments (*p* < 0.001) were predictors of worse physical function.

In the active disease group, breast (40.9; 95% CI 37.3 to 44.4), gynecological (40.7; 95% CI 34.16 to 47.2), and colorectal cancer (40.5; 95% CI 35.0 to 46.0) had the highest LS means (Figure 1). Brain (29.8; 95% CI 24.5 to 35.1), lymphoma (32.8; 95% CI 26.6 to 39.0), sarcoma (33.3; 95% CI 28.3 to 38.3), and prostate cancers (35.5; 95% CI 30.2 to 40.9) had the lowest LS means. Statistically significant differences (Appendix A) were seen between the cancers with higher means compared with those with lower means (e.g., a significant difference was found between colorectal cancer and sarcoma (*p* < 0.032) and colorectal cancer and brain tumor (*p* < 0.003)). In the non-active cancer group, sarcoma (40.6, 95% CI 36.7, 44.5) had the highest LS mean and colorectal cancer (35.9, 95% CI 30.3 to 41.6) had the lowest LS mean.

### 3.3. Social Participation

A significant differential effect was noted between type of cancer and cancer activity status for the social participation scale (*p* = 0.038). The model with the additional covariates showed marginal significance (at *p* < 0.10) for social participation (*p* = 0.079). Because of relative agreement between the models, subset analysis was performed to explore any differential effect. Within the active cancer subgroup analysis, social participation was associated with cancer type (*p* = 0.008). There was no association between social participation and cancer type in the non-active disease group (*p* = 0.708). Lymphoma without allogeneic BMT (34.8; 95% CI 28.5 to 41.2) had the worst social participation LS mean compared to individuals with breast cancer (*p* = 0.020). Increased age (*p* = 0.024), lower BMI (*p* = 0.002), and the presence of moderate or severe neurological impairment (*p* = 0.016) were associated with worse social participation in the non-active disease group. Like physical function, there were significant pairwise differences detected (beyond differences with breast cancer) in the LS means for cancer types in the group with active disease (Figure 1).

### 3.4. Fatigue

There was no interaction detected between type of cancer and cancer activity status (*p* = 0.383); thus, the whole sample was used in the multivariable analysis. A multivariable analysis evaluating the fatigue scale showed there was an association with cancer disease status (*p* < 0.001), with active cancer status predicting higher fatigue scores. Decreased age (*p* = 0.027), increased BMI (*p* < 0.001), and presence of premorbid musculoskeletal impairment (*p* = 0.015) predicted increased fatigue.

## 4. Discussion

This study evaluated the differences in patient-reported physical function, fatigue, and social participation between cancer types, cancer status, and patient characteristics, using the PROMIS Cancer Function Brief 3D Profile. Overall, the results showed that when individuals have active disease, cancer type does independently impact physical function and social participation. Fatigue was not impacted by cancer type, but was similarly high in all cancer types, especially if the cancer was active. In non-active disease, cancer type does not impact function, but age, BMI, and premorbid impairments are greater predictors of reduced physical function.

This is the first study of its kind to evaluate multidomain function across multiple disease groups and considering disease-specific and clinical factors that may impact function. Often, studies looking at symptomology in cancer patients (including reduced function) are limited to one disease type and institution; our study looks across all of the primary cancers seen in rehabilitation clinics, and the multicenter nature of this study makes the results widely applicable across practice locations. The results may inform referral pathways to cancer rehabilitation services—for example, older patients with a high BMI diagnosed with sarcoma may be flagged for rehabilitation services earlier in their treatment course.

For physical function, individuals with active brain tumors and lymphoma without a history of allogeneic bone marrow transplant had the lowest reported mean physical and social function scores. It is not surprising that patients with primary brain tumors had low function due to the multitude of neurologic deficits a space-occupying intracranial tumor can cause. Common symptoms from brain tumors include headache, paresis, seizures, ataxia, cognitive changes, bowel and bladder dysfunction, visual impairments, dysphagia, and dysarthria [14]. In general, brain and other nervous system cancer are associated with significant morbidity and mortality, with a 5-year relative survival of 33.8% [15]. Low physical function was also noted among patients with lymphoma. Lymphoma can cause considerable disability from bony and neurologic involvement of the disease, particularly when extranodal sites (e.g., central nervous system and spine) are involved [16]. These symptoms are often a late manifestation of disseminated disease. Chemotherapy for lymphoma may also cause peripheral neuropathy [17], further affecting function. However, the 5-year relative survival for lymphoma is relatively high (74–85%) [15], so selection bias may have played a role in our findings as the patients in our sample likely had more aggressive/advanced lymphoma requiring rehabilitation management.

Patients with breast, gynecological, and colorectal cancer had the highest reported mean physical function scores in patients with active disease. These patients likely have a relatively high global function but live with specific impairments that cause pain and restriction (e.g., post-mastectomy pain syndrome, glenohumeral adhesive capsulitis, lymphedema). While these impairments are bothersome and somewhat limiting, these patients are usually still able to perform their activities of daily living (ADLs) and instrumental activities of daily living (iADLs) relatively independently.

Patients with cancer have been shown to have worse physical and mental quality of life compared with individuals without cancer, which can impact their ability to participate in social roles and activities [18]. Our findings add to this evidence base by showing that specific cancer types impact social participation among those with active disease. Lymphoma had the lowest mean social participation score compared with all other cancers evaluated. This may reflect the chronically reduced function and side effects of systemic therapy to treat the disease. Survivors of more aggressive lymphoma subtypes may be less participatory than survivors of less aggressive lymphoma subtypes (e.g., Hodgkin lymphoma), who tend to be younger and potentially receive less toxic treatment [19]. Colorectal cancer had the highest mean social participation score, which may be secondary to early screening in our study population. Colorectal cancer is most frequently diagnosed between 65–74 years old [20], but the mean age of our sample was 56 years old, potentially correlating with earlier disease. When treated in its early stages, treatment generally consists of complete surgical resection through colonoscopy or laparoscopic surgeries that typically have short recovery periods, thus minimally impacting social participation or physical activity.

Cancer-related fatigue is highly prevalent in patients with cancer before, during, and after treatment [5]. Our results found fatigue to be associated with active disease status; specifically, active cancer was associated with higher fatigue by four points on the fatigue scale compared with non-active cancer. Fatigue was highest among those with lymphoma and gynecological cancers; however, there were no statistically significant differences noted among the cancer types. A retrospective cohort study evaluating the symptom burden of hematological malignancies found fatigue to be the most troublesome in the palliative setting [21]. Cancer-related fatigue has also been reported to be one of the most common side effects among individuals with gynecological cancers [22]. Fatigue can significantly affect health-related quality of life because it limits the ability to stay active and participate in ADLs and iADLs. It is not unexpected that fatigue is higher during active disease, as the treatments they undergo (e.g., chemotherapy, radiation, surgery) contribute to fatigue.

The additional covariates of age, BMI, and premorbid neurological and musculoskeletal impairments were also evaluated to see if any of these independently predicted functional outcomes. These variables most significantly impacted physical function and social participation among those with non-active cancer. However, they did not significantly predict function among individuals with active disease. The one exception was that higher BMI appeared to predict lower physical function in active disease. This may be because increased BMI often correlates with obesity. Obesity is a known risk factor for certain cancers [23,24] and can contribute to reduced function [25,26]. Cancer type did predict physical function and social participation in active disease but did not predict function in non-active disease. This could potentially be explained by the different approaches to treatment for each cancer type (e.g., surgery, radiation therapy, chemotherapy, bone marrow transplant, immunotherapy, hormone therapy, targeted drug therapy, cryoablation).

The changes in physical function, social participation, and fatigue can prevent individuals with cancer from performing ADLs/iADLs [4] or returning to work [27,28], cause considerable caregiver burden [29], and reduce quality of life [30]. Because these functional impairments can lead to greater morbidity and mortality, our findings are in alignment with the literature [31] supporting the incorporation of impairment-driven rehabilitation into the cancer care continuum. Our findings support treatment of functional impairments involving an interdisciplinary team of rehabilitation providers including physiatrists, physical therapists, occupational therapists, speech-language pathologists, rehabilitation psychologists, recreational therapists, and other supportive care services that serve to optimize an individual’s function and quality of life.

This study is novel for several reasons, and the results may inform clinical practice and future research. First, the findings from this study help characterize some of the main determinants of functional disability that may occur in specific cancer types. By understanding which groups of patients are more susceptible to certain functional impairments, early screening and monitoring for the presence or development of these impairments can be included into regular cancer care, and rehabilitation services can potentially be involved earlier and more frequently. Furthermore, specific education and counseling can be given to patients and their families on what to anticipate after cancer diagnosis and treatment, and the risk for lasting functional impairments.

Regarding future directions for research, the results may be helpful in developing better prognostic models and guiding individualized impairment-driven medical care among different cancer populations. With these data, patients may be identified for rehabilitation care, allowing for quicker and earlier intervention, potentially reducing or preventing morbidity.

## 5. Limitations

This study has limitations inherent to retrospective research, including incomplete capture of data and/or documentation. The patients with cancer evaluated in this study received cancer rehabilitation care and were more likely to represent those with more significant impairments or disabilities, rather than those patients not receiving cancer rehabilitation care. Cancer stage was not evaluated so it is uncertain to what extent this may have played in the participants function. The additive role of treatment determinants was also not assessed due to the large variation of treatments between cancer types, but it certainly could have a role in functional outcomes. For example, one study on breast cancer found that the presence of brain metastases, chemotherapy-induced peripheral neuropathy, and age impacted function [32]. Future studies evaluating the impact of specific treatment modalities on functional outcomes in active cancers would be beneficial.

## 6. Conclusions

These findings suggest that specific cancers may impact physical function and social participation differently during the active disease phase. Fatigue is experienced to a greater extent with active disease and is not significantly different between cancer types. Anticipatory guidance and education on the common impairments observed in specific cancer types and during certain disease stages may help improve functional outcomes. For long-term survivors without active disease, patients at risk of low function due to factors such as age, BMI, and non-cancer musculoskeletal or neurologic impairment should be screened for rehabilitation referral.

## Figures and Tables

**Figure 1 cancers-15-03515-f001:**
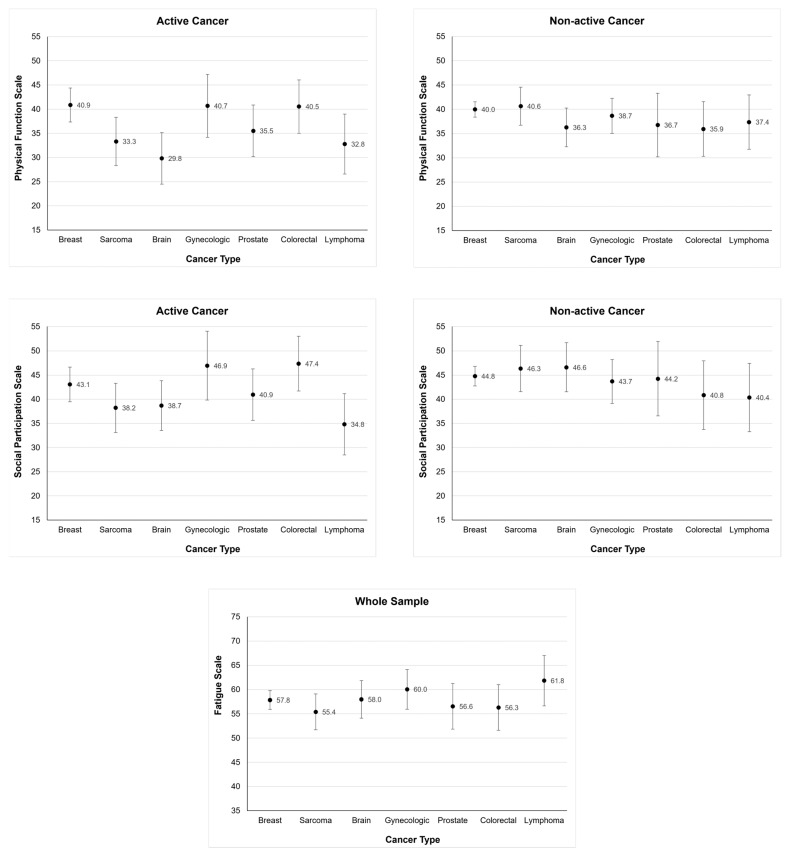
Least squares mean function across cancer types and functional domains.

**Table 1 cancers-15-03515-t001:** Demographics and baseline characteristics of the study sample by cancer type.

Characteristic	Overall (*n* = 332)	Breast (*n* = 209)	Sarcoma (*n* = 28)	Brain (*n* = 26)	Gynecologic (*n* = 23)	Prostate (*n* = 19)	Colorectal (*n* = 15)	Lymphoma (*n* = 12)	*p*-Value
Age (yrs)									<0.001
Mean (SD)	56.4 (13.4)	56.1 (11.6)	47.9 (15.3)	50.2 (17.4)	58.3 (11.6)	73.8 (8.6)	59.4 (12.5)	60.6 (15.1)	
BMI, *n* = 330									0.827
Mean (SD)	3.3 (1.2)	3.3 (1.2)	3.2 (1.3)	3.2 (1.3)	3.1 (1.2)	3.5 (1.3)	3.7 (1.5)	3.4 (1.1)	
BMI, *n* (%)									
<18.5	2 (1)	1 (0)	0 (0)	0 (0)	1 (4)	0 (0)	0 (0)	0 (0)	
18.5–25	104 (32)	67 (32)	10 (36)	10 (38)	6 (26)	4 (21)	4 (27)	3 (25)	
>23–30	104 (32)	59 (29)	10 (36)	9 (35)	9 (39)	9 (47)	5 (33)	3 (25)	
>30–35	66 (20)	48 (23)	4 (14)	3 (12)	4 (17)	2 (11)	1 (7)	4 (33)	
>35–40	27 (8)	18 (9)	1 (4)	1 (4)	2 (9)	1 (5)	2 (13)	2 (17)	
>40–45	27 (8)	14 (7)	3 (11)	3 (12)	1 (4)	3 (16)	3 (20)	0 (0)	
Gender *, *n* (%)									0.179
Male	60 (18)	3 (1)	15 (54)	11 (42)	0 (0)	19 (100)	4 (27)	8 (67)	
Female	272 (82)	206 (99)	13 (46)	15 (58)	23 (100)	0 (0)	11 (73)	4 (33)	
Active Disease, *n* (%)									0.004
Non-active	209 (63)	146 (70)	15 (54)	14 (54)	16 (70)	6 (32)	6 (40)	6 (50)	
Active	123 (37)	63 (30)	13 (46)	12 (46)	7 (30)	13 (68)	9 (60)	6 (50)	
Neurological impairment, *n* (%)									0.170
None/Minimal	298 (90)	190 (91)	26 (93)	23 (88)	21 (91)	18 (95)	12 (80)	8 (67)	
Moderate/Severe	34 (10)	19 (9)	2 (7)	3 (12)	2 (9)	1 (5)	3 (20)	4 (33)	
Musculoskeletal impairment, *n* (%)									0.241
None/Minimal	224 (67)	137 (66)	21 (75)	23 (88)	14 (61)	12 (63)	9 (60)	8 (67)	
Moderate/Severe	108 (33)	72 (34)	7 (25)	3 (12)	9 (39)	7 (37)	6 (40)	4 (33)	

Association between cancer type and Age & BMI were tested with ANOVAs. Associations between cancer type and categorical measures were tested with Fisher’s exact tests. * Gender-specific cancers (gynecologic, prostate, and breast cancers) were excluded from the Fisher’s exact test.

**Table 2 cancers-15-03515-t002:** Multivariable analysis of physical function, social participation, and fatigue.

	Physical Function				Social Participation				Fatigue	
	Active (*n* = 118)		Non-Active (*n* = 201)		Active (*n* = 122)		Non-Active (*n* = 203)		Whole Sample (*n* = 327)	
Variable	Estimates (95% CI)	*p* Value	Estimates (95% CI)	*p* Value	Estimates (95% CI)	*p* Value	Estimates (95% CI)	*p* Value	Estimates (95% CI)	*p* Value
Intercept	47.41 (39.71, 55.10)		47.37 (42.23, 52.51)		45.40 (37.84, 52.97)		46.99 (40.53, 53.45)		52.51 (47.30, 57.72)	
Cancer type (Ref: Breast)	Overall Effect	** *<0.001* **		0.340		** *0.008* **		0.708		0.379
Gynecologic	−0.18 (−6.18, 5.81)	0.952	−1.30 (−4.93, 2.32)	0.479	3.87 (−2.72, 10.46)	0.247	−1.10 (−5.66, 3.46)	0.636	2.19 (−1.78, 6.17)	0.279
Colorectal	−0.33 (−5.76, 5.10)	0.905	−4.03 (−9.79, 1.72)	0.168	4.29 (−1.26, 9.84)	0.128	−3.94 (−11.19, 3.30)	0.284	−1.55 (−6.35, 3.26)	0.526
Lymphoma, no alloBMT	−8.08 (−14.83, −1.32)	** *0.020* **	−2.61 (−8.35, 3.12)	0.370	−8.25 (−15.15, −1.35)	** *0.020* **	−4.42 (−11.64, 2.81)	0.229	3.99 (−1.36, 9.34)	0.144
Brain	−11.05 (−16.18, −5.92)	** *<0.001* **	−3.69 (−7.74, 0.35)	0.073	−4.39 (−9.25, 0.46)	0.076	1.82 (−3.34, 6.98)	0.488	0.13 (−3.68, 3.94)	0.946
Sarcoma	−7.55 (−12.18, −2.92)	** *0.002* **	0.66 (−3.24, 4.57)	0.738	−4.86 (−9.59, −0.12)	** *0.044* **	1.56 (−3.19, 6.32)	0.517	−2.45 (−6.10, 1.19)	0.186
Prostate	−5.35 (−10.54, −0.16)	** *0.043* **	−3.22 (−9.68, 3.24)	0.326	−2.13 (−7.27, 3.02)	0.414	−0.55 (−8.09, 6.99)	0.885	−1.29 (−5.92, 3.35)	0.586
Age	−0.02 (−0.14, 0.09)	0.710	0.05 (−0.03, 0.13)	0.248	0.01 (−0.10, 0.13)	0.848	0.12 (0.02, 0.23)	** *0.024* **	−0.09 (−0.18, −0.01)	**0.027**
BMI	−1.19 (−2.38, −0.01)	** *0.047* **	−1.68 (−2.47, −0.89)	** *<0.001* **	−0.35 (−1.55, 0.84)	0.557	−1.62 (−2.62, −0.62)	** *0.002* **	1.68 (0.88, 2.48)	** *<0.001* **
Neurological impairment	−1.72 (−8.05, 4.61)	0.591	−4.76 (−7.72, −1.81)	** *0.002* **	−1.60 (−8.06, 4.85)	0.624	−4.64 (−8.41, −0.88)	** *0.016* **	3.27 (−0.15, 6.69)	0.061
Musculoskeletal impairment	−1.15 (−4.43, 2.12)	0.487	−4.36 (−6.62, −2.11)	** *<0.001* **	−2.06 (−5.35, 1.22)	0.215	−2.58 (−5.39, 0.23)	0.072	2.83 (0.56, 5.09)	** *0.015* **
Active cancer(Ref: Non-active)	-------		-------		-------		-------		4.14 (2.02, 6.26)	** *<0.001* **

Bold/italics indicates statistical significance (*p* < 0.05). Premorbid neurological and musculoskeletal impairments compare moderate or severe impairments to none/minimal.

## Data Availability

The data presented in this study are available on request from the corresponding author.

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
