# Peer review of "Function in Cancer Patients: Disease and Clinical Determinants"

_cancers, 2023, doi:10.3390/cancers15133515_

Round 1

Reviewer 1 Report

Partial cognitive and social impairment is a well-known after-effect of various cancer types during the treatment and post treatment regime. However, the statistical data that are available to understand the correlations between different cancer types and these aftereffects are insufficient. In the current article, the authors have discussed how specific cancers may impact physical function and social participation differently during the active disease phase. The study was conducted between 332 patients with various types of cancers at different stages of progression or recovery. Moreover, the authors also studied the relationship of fatigue with different cancer types at different stages. The article also discussed the statistical correlations between the age of different patients with different cancer types that affect the said aftereffect. In this context, the current article is important to the scientific community. The data were well discussed in with all the statistical analysis and the limitations of the study was also duly discussed. Hence, I recommend the article to be published after the following minor revisions:

1. There is no description and legends for Figure 1. Please add due description of the figure 1 at the bottom of the figure.

Partha

English Language is fine. Minor spell check needs to be done.

Author Response

Thank you for your helpful comments, and for taking the time to review our manuscript. We greatly appreciate it. To your suggestion, we added a description/legend for Figure 1.

Reviewer 2 Report

This study is interesting and my favorite:

The introduction of the article should be improved.

The discussion of the article should be improved and more articles should be compared.

Check the article for grammar.

The way of referencing should be observed.

Author Response

Thank you for your helpful comments, and for taking the time to review our manuscript. We greatly appreciate it.

The introduction of the article should be improved.

-Response: Thank you for the suggestion – we added to the second paragraph to describe the importance of our study, and existing evidence gaps.

The discussion of the article should be improved and more articles should be compared.

-Response: Thank you; we have enhanced the discussion and made more clear the results and why they are important. We are interested in hearing your thoughts about it.

Check the article for grammar.

-Response: Thank you; we have edited it again for grammar

The way of referencing should be observed.

-Response: Thank you; we have edited the references to correct the error.

Reviewer 3 Report

In this manuscript, the authors try to evaluate differences in physical function, fatigue, and social participation in total of 332 patients of breast cancer, sarcoma, primary brain, gynecologic, prostate, colorectal and lymphoma. They found that when individuals have active disease, cancer type does independently impact physical function and social participation. Fatigue was not impacted by cancer type, but was similarly high in all cancer types, especially if the cancer is active.

Major issue:

1. The title is too general to represent the research which this manuscript is involved.

2. What’s the novelty of this manuscript? Can the author highlight the difference of their finding to previous publication?

3. What’s the summary conclusion of physical function, fatigue and social participation? The authors need to indicate the conclusion in the sub-title of result section.

4. How can we utilize the finding of this manuscript mentioned for cancer rehabilitation clinics? The authors need to elucidate this question in detail.

Minor issue:

There is a redundant line break in the last sentence of the abstract.

Author Response

Thank you for your helpful comments, and for taking the time to review our manuscript. We greatly appreciate it.

Major issue:

  1. The title is too general to represent the research which this manuscript is involved.

 - Response: we edited the title to be more clear, thank you for the suggestion.

  1. What’s the novelty of this manuscript? Can the author highlight the difference of their finding to previous publication?

 - Response: Thank you, we do need to highlight this more. In addition to enhancing the introduction to explain why this study was necessary, we have added to the discussion section (second paragraph, last paragraph) to highlight the novelty and utility of this study.

  1. What’s the summary conclusion of physical function, fatigue and social participation? The authors need to indicate the conclusion in the sub-title of result section.

 - Response: Thank you for this suggestion; we agree it could have been made clear. We now state the results in the beginning of the results section; we then go into detail in the ensuing paragraphs.

  1. How can we utilize the finding of this manuscript mentioned for cancer rehabilitation clinics? The authors need to elucidate this question in detail.

- Response: Thank you; we now provide an example in the second paragraph of the Discussion setting, as well as the last paragraph.

Minor issue:

There is a redundant line break in the last sentence of the abstract.

-Response: We have fixed this, thank you.

Round 2

Reviewer 3 Report

No further comments.